# Image Restoration for Training Data Reconstructed from Trained Neural Networks

## Abstract

Haim et al. [NeurIPS 2022] propose a method to reconstruct training data from trained neural networks with impressive results. While their reconstructed images resemble the original training images, most of them also contain a considerable amount of noise and artifacts. This is especially true, when the network was trained on more than just a few dozen images. To address this, we view the problem as a specific image restoration task. Since the noise and artifacts are different from other types of noise (Gaussian noise, compression artifacts, blurring, or impulse noise from digital cameras), we create a new dataset specifically for the restoration of images produced by the reconstruction process proposed by Haim et al. We use this dataset consisting of about 60 million noisy reconstructions of CIFAR-10 images to train a diffusion model on the restoration task. Using this method, we obtain reconstructions that are significantly closer to the original training images measured in terms of SSIM and HaarPSI scores.

## 1 Introduction

To what extent is the training data encoded in the weights of a trained neural network? This is not just an important question for our understanding of neural networks, but also a question with potential privacy implications. Since the work of Fredrikson et al. (2014), model inversion attacks have been investigated for many years. Fredrikson et al. (2015) for example show that, given a facial recognition model, it is possible to obtain an image depicting a target person. This is done by optimizing for an input of a facial recognition model that maximizes the models certainty that the input shows a particular target person. However, such model inversion attacks fail to produce any meaningful result if a class contains many diverse images that are not very similar to one another (Shokri et al., 2017).

Haim et al. (2022) develop another type of attack that can recover part of the training data of an image classifier even when each class contains a diverse set of images. They are able to recover a number of training images from the weights of a given trained network. The reconstructed images can be noisy, but clearly resemble specific individual images from the training set (as opposed to simply constructing new realistic images that would belong to a class, but were not in fact used during the training).

We view the removal of noise and artifacts from the reconstructed images produced by Haim et al. (2022) as a new type of image restoration task and by tackling this new task, we significantly improve the reconstructions obtained by Haim et al. (2022). We create a large dataset consisting of such noisy reconstructions together with their clean counterparts from the original training set and then train a conditional diffusion model on this new image restoration task. This works even when the corruption of the images is severe and even when it would be difficult for a human to interpret the corrupted image.

Previous works have considered using generative tools to guide model inversion attacks. The idea is that an unconstrained optimization over the model input to maximize a certain model output may lead to completely unrealistic results. Instead, the goal is to either constrain the generation or bias it towards specific types of "realistic" inputs. Zhang et al. (2020) use pretrained GANs in an attempt to prevent the model inversion attack from generating unrealistic images. However, while this approach generates realistic images that fit a particular class, just as the classic unguided model inversion attack, it cannot recover images used in training if the images in the class are sufficiently diverse. In this sense our results are notably different. While our use of a diffusion model also risks

hallucinations—the production of images that look realistic, but were in fact not part of the training data—we propose a mitigation for this that utilizes the probabilistic nature of the diffusion process.

Our main contributions can be summarized as follows:

- We demonstrate that diffusion models can be very effective at removing noise and artifacts from reconstructed training data, specifically reconstructions that are produced by solving the optimization problem based on Karush–Kuhn–Tucker(KKT) points of the maximum-margin problem.

- As far as we are aware, this is the first work that successfully trains and applies a diffusion model for improving imperfect results of solving a specialized optimization problem, where the imperfections appear to be a special kind of noise not otherwise familiar and for which off-the-shelf denoisers do not appear suitable for.

- We contribute a dataset consisting of about 60 million pairs of reconstructed images of varying quality and their estimated (augmented) CIFAR-10 counterpart for the community to use in future research.

## 1.1 RELATED WORK

We refer to Haim et al. (2022) for a detailed comparison between their approach and other types of attacks.

Buzaglo et al. (2023) extend the work of Haim et al. (2022) in a number of ways. Firstly, they show that the approach can also reconstruct part of the training data in the multiclass setting. In fact, they demonstrate that trained models become more vulnerable to training sample reconstruction as the number of classes increases. They also study regression loss functions with weight decay (as opposed to cross entropy loss) and establish that reconstructions of training samples is still possible. Finally, they also successfully perform reconstructions for models trained on a larger training set (size 5000 as opposed to the previous largest of 1000 in Haim et al. (2022)).

Building on Haim et al. (2022) and model inversion methods such as that of Tumanyan et al. (2022), more recently Oz et al. (2024) show how one may reconstruct higher-resolution training data from fine-tuned foundation models via the embedding space.

Feldman (2020) provide a theoretical model demonstrating that for natural data distributions memorization of labels is necessary for achieving close-to-optimal generalization error. Hayes et al. (2023) obtain a tight upper bound on the success of any reconstruction attack against DP-SGD, a standard algorithm for private deep learning.

Balle et al. (2022) show that it is feasible for an adversary who knows all the training data points except one to reconstruct the remaining data point.

Jagielski et al. (2023) show that, although non-convex models can memorize data forever in the worst-case, standard image, speech, and language models empirically forget examples as training time increases. Carlini et al. (2023b) identify three log-linear relationships that quantify the degree to which large language models emit memorized training data.

Carlini et al. (2023a) devise a generate-and-filter pipeline, and show that it is able to extract over a thousand training examples from state-of-the-art diffusion models. Somepalli et al. (2023) study how to detect reproduction of training images by diffusion models, and investigate how its rates are impacted by factors such as training set size.

## 1.2 PRELIMINARIES

The approach by Haim et al. (2022) relies on the implicit bias of gradient descent. Specifically, Lyu & Li (2020) and Ji & Telgarsky (2020) show that when optimizing the binary cross-entropy loss of a homogeneous neural network using gradient flow, under certain conditions, the direction of the network weights converges to a Karush–Kuhn–Tucker (KKT) point of the maximum-margin problem. Concretely, suppose the training data consists of $n$ labeled examples $(x_i, y_i)$ with $y_i \in \{-1, 1\}$ for $i \in [n]$ and that at some point during training, we have $\min_i\{y_i\Phi(\theta; x_i)\} \geq 1$, where $\Phi(\theta; x)$ is the output of the neural network with weights $\theta$ on input $x$. This condition $\min_i\{y_i\Phi(\theta; x_i)\} \geq 1$

mainly means that we require that all training samples are classified correctly. Then, there exist non-negative $\lambda_i$ such that in the limit, as the training goes to infinity, the direction of the weights $\theta/\|\theta\|$ converges to $\theta^*$ with

$$\theta^* = \sum_{i=1}^{n} \lambda_i y_i \nabla \Phi(\theta^*; x_i) \,. \tag{1}$$

Additionally, for all $i \in [n]$ for which $y_i \Phi(\theta^*; x_i) > \min_i\{y_i \Phi(\theta^*; x_i)\}$ (that is for all training samples that do not lie on the margin), we will have $\lambda_i = 0$.

Therefore, if we are given weights of a network that has been trained for a long time, we can attempt to (approximately) solve eq. (1) for $\lambda_i$, $x_i$, and $y_i$ to recover the training samples. Note that this is not suitable to recover training samples that do not lie on the margin, because for those $\lambda_i = 0$.

We do not need to know the number $n$ of samples in the training data, but only an upper bound $m$ on the number of training samples that are in the same class and lie on the margin. We then aim to minimize the loss

$$L(\lambda', x') = \left\|\theta^* - \sum_{i=1}^{2m} \lambda_i' y_i' \nabla \Phi(\theta^*; x_i')\right\|_2^2$$

over $\lambda'$ and $x'$, where $y_i'$ is fixed to $-1$ for $i \leq m$ and $y_i'$ is fixed to $1$ for $i > m$. In principle, this would allow for the reconstruction of up to $m$ training samples for each class, meaning we do not have to know the exact size of the training data nor the class distribution in the training data. This potential overestimate in training samples per class does not invalidate the approach because for any extra term in the sum it is always possible to set $\lambda_i' = 0$.

Rather than optimizing for the loss $L(\lambda', x')$ as written, Haim et al. introduce a few additional technical tweaks. Firstly, they substitute the gradient computation of the ReLU function by the derivative of the Softplus function $\ln(1 + e^{\alpha x})/\alpha$, with a hyperparameter $\alpha$. The hope is that this is easier to optimize due to it being continuous, but, especially for large $\alpha$, otherwise not very different from the derivative of the ReLU function. Secondly, they add terms to the loss function that penalize pixel values outside the range $[-1, 1]$ and values for $\lambda_i'$ smaller than a hyperparameter $\lambda_{\min}$. Concretely, they add a term of $\max(p - 1, 0)^2 + \max(-1 - p, 0)^2$ for each pixel $p$ of each $x_i$ to the loss and a term of $5 \cdot \max(-\lambda_i + \lambda_{\min}, 0)^2$ for each $\lambda_i$.

## 2 CREATION OF THE DATASET

The reconstructions of training data that are obtained by minimizing the loss function stated in section 1.2 contain considerable noise and artifacts. One reason for this is that in practice, networks are not trained to infinity and therefore eq. (1) is not exactly satisfied. We see the task of "cleaning up" such noisy reconstructions as a specific type of image restoration task.

In order to be able to train a model, in our case mainly a diffusion model, on our image restoration task, we have to create a suitable dataset for this particular task. When, say, training for an image restoration task that aims to remove Gaussian noise, it is possible to generate the corresponding image corruptions on the fly during training. For this, we only require a set of clean images, which we can then augment (through flips, rotations, etc.) and corrupt by adding Gaussian noise to obtain pairs of input images and the corresponding target image. In contrast, we cannot generate the type of corruption occurring in our restoration task on the fly in an efficient way. Instead, we create a dataset consisting of a large number of pairs of images. One image in each pair is an original image from the CIFAR-10 training set and the second image is a corrupted version of the same image resulting from the reconstruction process.

To generate such pairs, we first train a neural network on some images of the CIFAR-10 training set and then attempt to reconstructed the training images. If the reconstruction procedure results in some images that, while noisy, can be confidently matched to one of the images the network was trained on, this gives us a pair of noisy reconstructed image and a clean target image from the CIFAR-10 training set.

**The training phase.** We train a neural network with two hidden layers, without bias terms, consisting of 1000 neurons each and with ReLU activations. As Haim et al., the weights in the first layer are

initialized using Gaussians with standard deviation $10^{-4}$, and all other layers are initialized using Kaiming He initialization (He et al., 2015). The training is done for the binary task of distinguishing animals from vehicles in images and uses binary cross entropy loss. Then we run the reconstruction procedure. As Haim et al., we normalize the training data by averaging all training images and subtracting this average from all training images.

**The reconstruction phase.** For all $i$, we randomly initialize $\lambda'_i \sim \mathcal{U}[0,1)$ and $x'_i \sim \mathcal{N}(0, \sigma^2 I)$, where $\sigma$ is a hyperparameter. We set $m = 200$ (which means we have $2m = 400$ many $\lambda'_i$ and $x'_i$'s). We then optimize the loss function described in section 1.2. Note that, as mentioned before, during this reconstruction phase, we substitute the gradient computation of the ReLU function by the derivative of the Softplus function with the hyperparameter $\alpha$.

**Improving the data generation speed.** One of our main challenge is that training a network and then running the reconstruction procedure of Haim et al. takes a considerable amount of time. In the main setup of Haim et al., the network is trained on 500 images for one million epochs. Haim et al. then perform the reconstruction given this network. A single such reconstruction run alone takes about 90 minutes when using a single NVIDIA GeForce RTX 3080 Ti GPU and may only produce a few reconstructions of sufficient quality. Here sufficient quality means that we can match it to one of the images the network was trained on with some level of confidence.

For this reason, we modify the procedure in order to generate more data for our dataset in a shorter amount of time. In particular:

- We use single instead of double precision.
- For both the training of the network and the reconstruction procedure, we use the Adadelta optimizer (Zeiler, 2012) instead of gradient descent. The Adadelta learning rate is fixed at 1.
- We train the network on 300 images (150 for each class) instead of 500.
- We train the network for 50,000 epochs instead of one million epochs.
- Instead of setting $m$ equal to the number of images the network was trained on, We run the reconstruction only with $m = 200$ (i.e. we use $400$ $x'_i$ and $\lambda'_i$ values).
- We fix the hyperparmeters for the reconstruction run as $\alpha = 20$, $\lambda_{\min} = 0.5$, and $\sigma = 0.001$.
- We run the reconstruction for 11,900 epochs instead of 50,000.

With these changes, the combined training and reconstruction takes a few minutes rather than hours.

**Image augmentation.** As is often done when training generative models, we aim to increase the diversity in our training set through image augmentation. While this is often done on the fly while training a diffusion model, as explained above, this is not possible for us. The difficulty is the following: suppose we have an augmentation operator $A$ that transforms a training image $I$ to an augmented image $A(I)$. Now suppose we get a noisy reconstruction $R$ of $I$ and the pair $(R, I)$ is stored in our dataset. Can we now train the diffusion network on an augmented version of $I$? For this we would want to obtain a reconstruction of $A(I)$. However, it is not obvious what such a reconstruction would be. One guess would be $A(R)$, but it is not clear to us that, say, rotating a training image would result in the same kind of rotation in the reconstructions with no other changes to the distribution of the reconstruction. Investigating this would be interesting future work, but for this current work, we are choosing the safe option of performing the data augmentation already when creating our dataset.

Concretely, after sampling 300 images from the CIFAR-10 training set (150 from "vehicle" classes and 150 from "animal" classes), we randomly augment each of the images using the same augmentation pipeline and parameters as Karras et al. (2022) (which in turn borrow the pipeline from Karras et al. (2020)). It randomly applies x-flips, y-flips, isotropic and anisotropic scalings, and fractional rotations and translations. To stop the augmentations from leaking into the image generation when applying the trained diffusion model, the augmentation parameters are given to the diffusion model as an additional conditioning input. When the model is used to generate new images, the augmentation parameters are set to 0 to condition the model to only generate non-augmented images. All this is done exactly as in Karras et al. (2022) and we refer to their work for more details.

**Identifying image pairs.** During each reconstruction, we save $x_i'$ at epochs 100, 200, 300, 500, 800, 1400, 2400, 4100, 7000, 10000, and 11900. Then for each of the $x_i'$ at each of these points in time, we compute the SSIM score (Wang et al., 2004) to each of the images the network was trained on. If there is an $x_i'$ at any of these times that has an SSIM score above 0.4 for some training image $T$, but where the SSIM score is at least 0.05 lower for the next best training image, then we say that we are reasonably confident that this $x_i'$ represents a corrupted version of $T$. We therefore add $x_i'$ and $T$ to our dataset. In fact, in this case we add all versions of $x_i'$ recorded in epochs $100, 200, \ldots, 11900$ even if only one of them produced a confident match with $T$. By taking even versions of $x_i'$ from very early epochs, we hope to also include image pairs in our dataset where one image is a corrupted version of the other, but where the corruption is so severe that the SSIM score is possibly even below 0.4.

**Number of runs, dataset size, and compute.** We ran this process of training a network and then performing the reconstruction 100k times; each time with a new random subset of 300 images from the full CIFAR-10 training set (but such that the classes are balanced) and their random augmentations. Unfortunately, some images are much more likely to be reconstructed than others, which creates big imbalances in the dataset and missing out on including many images of the CIFAR-10 training set completely because they were not reconstructed with suitable quality even once in any of the 100k runs. To mitigate this, in a second phase, we ran the process another 50k times, but this time only sampling from the 4000 CIFAR-10 images least seen so far in the dataset rather than from all 50,000 images in the CIFAR-10 training set (which images were least seen is updated after each run). Indeed, while imbalances remain, with this approach we were able to ensure that 42,707 of the 50,000 images in the CIFAR-10 training set appear at least once in the dataset.

The total compute for generating the dataset on an internal cluster utilizing NVIDIA A100 GPUs was about 5600 GPU hours. We believe there is room to improve this by using a better sampling strategy when picking images to train the network on. This could make better use of our new gained knowledge about the discrepancies in likelihood of different images having good enough reconstructions in a single run. For example, it may be advisable to avoid repeatedly sampling images for which already many suitable corrupted versions were generated. At the same time, it may also help to not too aggressively sample from the images that have repeatedly failed to produce any suitable reconstructions over a large number of runs. In our experiments, we saw that otherwise there can be many runs not contributing a single new entry to the dataset because all images used in training the network were chosen from the set of images that are unlikely to produce meaningful reconstructions.

## 3    THE CONDITIONAL DIFFUSION MODEL

For our diffusion model, we use the DDPM++ architecture (Song et al., 2021) and exactly the same method (preconditioning, noise distribution, etc.) as Karras et al. (2022). We only make two notable changes to their code (which is available[1] under a CC BY-NC-SA 4.0 license).

The purpose of the first change is to condition the model on a given (corrupted) image that is supposed to be restored. For this we use six instead of three input channels, where the three colors of the corrupted image are fed into the network through the three additional input channels. The skip connection in the preconditioner is still restricted to the first three channels, however.

The second change is to use our dataset consisting of pairs of images from the CIFAR-10 training set and noisy reconstructions of those as described in section 2. This also means data augmentation already has been performed when creating the dataset rather than being done on the fly at the time of training the diffusion model. We emphasize that it is important that our dataset used to train the diffusion model only relies on the training set of CIFAR-10, such that we can later evaluate it using images involving the test set that the diffusion model has not seen during training.

**Dataset balancing.** Our dataset containing about 60 million examples contains augmented versions of some CIFAR-10 images much more frequently than others. We balance this as follows: if a CIFAR-10 image appears $z$ times within the dataset, we remove each such occurrence from the

---

[1]https://github.com/NVlabs/edm/tree/62072d2612c7da05165d6233d13d17d71f213fee

dataset with probability $1 - 11/z$, independently (every CIFAR-10 image that appears in the dataset, appears at least eleven times). After this step, in expectation, each image appears eleven times and we expect to have around 460k image pairs. We repeat this process about 140 times to obtain 140 different random balanced subsets of our full dataset. We then combine all of these subsets together into a new large dataset containing about 64 million examples and that is balanced in expectation.[2] (Some examples in this new dataset are repeated several times, but the key is that no CIFAR-10 image is greatly over-represented.)

**Adding random noise examples.**    The first time we save $x_i'$ values when collecting data for our dataset is after 100 epochs (see section 2). To improve the diversity of images produced by the diffusion model when conditioned on extremely noisy and ambiguous reconstructions, we add 10% of training samples consisting entirely of random noise as follows: we simulate what $x_i'$ values would be like just after initialization prior to any optimization taking place (i.e. at epoch 0). That is, we average 300 images of the CIFAR-10 dataset and add random noise to this average. This noisy "image" is then paired with one of the 300 images we averaged over. We exploit the fact that the diffusion model samples images from a wide distribution when conditioned on very unclear corrupted images to mitigate hallucinations. (see Figure 4).

**Training details and compute.**    The training was done in double precision, with batch size 513 and learning rate 0.001. The training, using three NVIDIA A100 GPUs, took about two days. However, around 20% of that time was spent on preparing data from the full dataset for the training.

## 4    THE CNN MODEL

Apart from conditional diffusion models, we can use our dataset to train other models as well. One simpler and less resource intensive model is a CNN. In particular, we train a model based on the CNN architecture introduced by Mohan et al. (2020). This architecture is based on DnCNN (Zhang et al., 2017), but without bias terms in any layer. We will refer to this as BF-DnCNN.

We trained the model for about 40,000 batches of size 128 using the Adam optimizer (Kingma & Ba, 2015) with initial learning rate 0.001 and then letting the learning rate half after the first 25,000 batches and further halving it every 5,000 batches after that. This is similar to what Mohan et al. (2020) propose.

We use this trained model as an additional baseline to compare our diffusion based approach to.

## 5    EVALUATION AND RESULTS

To evaluate how well the trained diffusion model performs in our specific image restoration task, we first replicate the reconstruction experiment of Haim et al. (2022) and then apply our diffusion model to the "noisy" reconstructions obtained.

### 5.1    REPLICATING THE RECONSTRUCTION EXPERIMENT

We train a neural network of the same architecture as described in section 2: three fully-connected layers with ReLU activations and where each hidden layer has 1000 neurons. The initalization is as described in section 2 and training is once again done using binary cross entropy loss for the task of distinguishing animals from vehicles.

Importantly, the network is trained on 500 images from the *test set* of ciFAIR-10 (Barz & Denzler, 2020). ciFAIR-10 is derived from CIFAR-10 by replacing duplicate images in the CIFAR-10 test set (images in the test set that appear in identical or near identical form in the CIFAR-10 training set or that appear more than once in the test set) with new images. Because our diffusion model has only been trained on images derived from the training set, this ensures that our diffusion model has not seen any of the 500 images we are now attempting to reconstruct.

---

[2]To save memory, we prepare data for the new balanced dataset on the fly while training the diffusion model rather than preparing it all upfront.

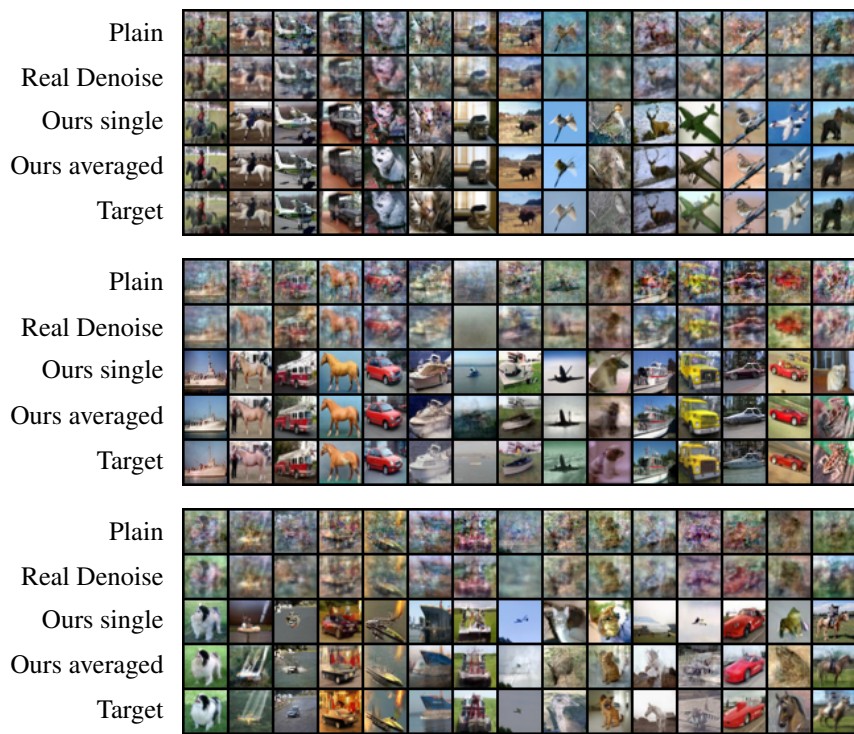

Figure 1: Among all training images, we show the 45 (in the bottom rows) that resulted in the best plain reconstruction using the method of Haim et al. (shown in the top rows) based on SSIM score. For each of these training images, we also show the best match (again based on SSIM score) after applying different image restoration methods to all reconstructed images. The second rows, show the result after applying the real image denoiser by Zamir et al. (2022). The third rows show the images achieving the highest SSIM score with the target after our image restoration is applied once to each plain reconstruction. The fourth rows show the images achieving the highest SSIM score with the target after our image restoration is applied 20 times and the results are averaged.

We train for one million epochs using gradient descent with learning rate 0.01 and full batch size. And once again, we normalize the inputs by averaging all images 500 and subtracting this average from all images as Haim et al. do.

The resulting trained network is the network from which we attempt to reconstruct the images it was trained on.

Haim et al. (2022) observed that their procedure has a number of hyperparameters and it is not necessarily clear how to choose them. They therefore repeat the reconstruction process a number of times, each time choosing a different initialization and different hyperparameters at random. They then combine all the results and identify the most successful reconstructions from that combined set of images.

For each of these reconstruction runs, for all $i$, we randomly initialize $\lambda_i' \sim \mathcal{U}[0, 1)$ and $x_i' \sim \mathcal{N}(0, \sigma^2 I)$, where $\sigma$ is a hyperparameter. We set $m = 500$ (which means we have $2m = 1000$ many $\lambda_i'$ and $x_i'$'s). We then use gradient descent to optimize the loss function described in section 1.2. Once again, during this optimization we substitute the gradient computation of the ReLU function by the derivative of the Softplus function with the hyperparameter $\alpha$.

We repeat the reconstruction 303 times with gradient decent with momentum 0.9 for 49,000 iterations and different hyperparameters: learning rate, $\sigma$, $\lambda_{\min}$, and $\alpha$. Specifically, the learning rate is chosen from a log-uniform distribution between 0.01 and 1, $\sigma$ is chosen from a log-uniform distribution between $10^{-6}$ and 0.1, $\lambda_{\min}$ is chosen uniformly from $[0.01, 0.5]$, and $\alpha$ is chosen uniformly from $[10, 500]$.

56 of the 303 runs resulted in low reconstruction quality after 5000 epochs and were aborted, leaving 247 successful runs. We record all attempted reconstructions after 10k, 15k, 25k, and 49k epochs of all successful 247 runs. Note that some of the $x_i'$ for some of the runs may reflect some training image, but many do not and seemingly just consist of unstructured noise.

**Postprocessing.** While the $x_i'$ resulting from a reconstruction run typically have pixel values not exceeding an absolute value of 1, due to the nature of the loss function in section 1.2, they are otherwise arbitrarily scaled (in particular, their range can be orders of magnitude smaller than the full interval $[-1, 1]$). We therefore normalize them by multiplying the pixel values with the largest possible constant $r$, such that after reversing the training data normalization (i.e., after adding the average of all images in the training data), the pixel values are in the interval $[0, 1]$.

### 5.1.1 EVALUATING OUR IMAGE RESTORATION PERFORMANCE

For each of the 500 images that the neural network was trained on, we find the best match in terms of SSIM score for [3] each of the following set of images:

1. The set of all images obtained in all of the reconstruction runs. We call these the plain reconstructions. They tend to be noisy and have artifacts.

2. The set of images obtained after applying the Restormer framework by Zamir et al. (2022) with pretrained weights for image deraining, Gaussian denoising, or real image denoising to all plain reconstructions.

3. The set of images obtained after applying the trained BF-DnCNN discussed in section 4 to all plain reconstructions.

4. The set of images obtained after applying our diffusion basaed image restoration model once to each plain reconstruction. By this we mean that we condition our trained diffusion model on the plain reconstruction and sample an image using 18 sampling steps.

5. The set of images obtained as follows: For each plain reconstruction, applying our image restoration model 20 times and taking the average of the 20 resulting images.

We are exploring the last method mainly for the purpose of mitigating hallucinations. Hallucinations can arise when the plain reconstruction we condition on is of very poor quality. In such cases, the diffusion model will produce vastly different outputs each time it is applied with different initial noise. Averaging such different outputs will not result in a clear image and be therefore less misleading than taking the output of only a single application of the diffusion model.

Figure 1 shows the results for the 45 original images the network was trained on that were reconstructed best (based on SSIM scores) for the *first* method. For each of those images, we then show the best reconstruction (again, based on SSIM scores) for the different methods. For the Restormer method, we only show the effect of real image denoising as a representative example. Gaussian denoising and image deraining performed worse (see Figure 2).

The plain reconstructions in the top rows of Figure 1 are of similar quality as the ones presented in Haim et al. (2022) for the comparable setting of reconstructing training images from a network that was trained on 500 images.

Note that even when only severely corrupted images are produced by the plain reconstruction procedure (to the point that the human eye can barely make out what the image shows), our image restoration is able to produce much clearer images that are very close to the original training image. However, the corruptions are very specific and distinct from common corruptions considered in image restoration. The image denoise and image derain procedures by Zamir et al. (2022) for example improve image quality only very slightly.

Figure 2 shows for each method how good its best reconstructions are. Specifically, it shows for the $i$th best reconstructed image, how accurately (based on SSIM for the left plot and HaarPSI

---

[3]For computational efficiency, as Haim et al. (2022), we only do this approximately. Instead of computing SSIM scores between each reconstructed image and each original training image, we initially match each reconstructed image to one original training image using a heuristic that is faster to compute than SSIM and then only compute the exact SSIM score between those matched images.

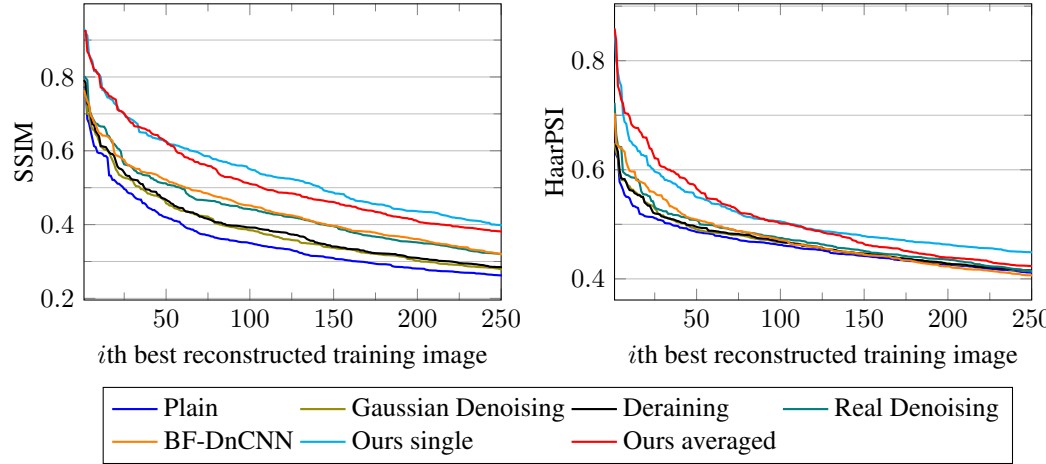

Figure 2: We plot for each method how good the best reconstructions are. Specifically, we plot for the $i$th best reconstructed image for a particular method, how accurately that image was reconstructed based on the SSIM and HaarPSI metric. Note that the $i$th best reconstructed image can be different for each method and also between the two similarity measures.

(Reisenhofer et al., 2018) for the right plot) that image was reconstructed. Note that the $i$th best reconstructed image can be different for each method and also between the two similarity measures.

Although in Figure 2 the $i$th best reconstructed image can be different for different methods, it is not the case that some training images have much better reconstructions after our image restoration method is applied, but others have much better matches among the plain reconstructions. Figure 3 shows that none of the 500 training images is reconstructed worse after our restoration procedure is applied to all plain reconstructions.

Buzaglo et al. (2023) suggest as a general guide that reconstructions with an SSIM score of above 0.4 to some training image should be considered a "good" reconstruction and they pay special attention to how many such good reconstructions can be obtained for a trained network. For our network, which was trained on 500 images, the plain method by Haim et al. results in 59 good reconstructions when using this definition. However, when we apply our image restoration method 20 times and averaging the outputs, this improves to 214 good reconstructions.

We observe that the Restormer archtiecture trained on different standard image restoration tasks gives slight, but not dramatic improvements compared to the plain reconstructions, with "Real Denoise" performing best among these. However, training a model specifically for the type of image restoration we want to perform achieves very good results. This suggests (possibly not surprisingly) that the type of corruption we are confronted with when reconstructing part of the training data of a neural network is fundamentally distinct from the noise we may encounter in images in other settings. Nevertheless, it is a specific type of corruption that can be effectively trained for.

## 6 LIMITATIONS AND FUTURE WORK

We demonstrate that it is possible to remove noise and artifacts from many images that are imperfect reconstruction of training images. We hope this paves the way to further extensions to higher resolution datasets, for example by also building on recent work on reconstructions of vectors in the latent space (Oz et al., 2024).

Another interesting direction to explore is whether the restoration task can still be learned across different distributions. Our diffusion model is trained on data that, while different from the data it is applied to, comes from the same distribution. Specifically, we train the diffusion model on data derived from the training set of CIFAR-10 and then evaluate the method using images from the test set of ciFAIR-10. Can similar results be achieved if there is a larger difference in the distributions? For

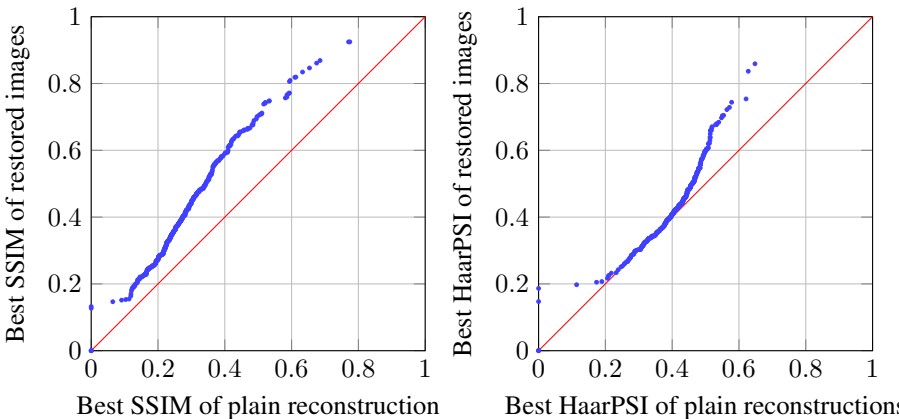

Figure 3: (left) A scatter plot showing for each of the 500 training images, what the SSIM score of the best reconstruction is before and after restoration. Here we use the restoration method that takes the average of 20 images generated by repeatedly applying our diffusion model. (right) A scatter plot for the same images as in the left plot, but evaluated using the HaarPSI metric.

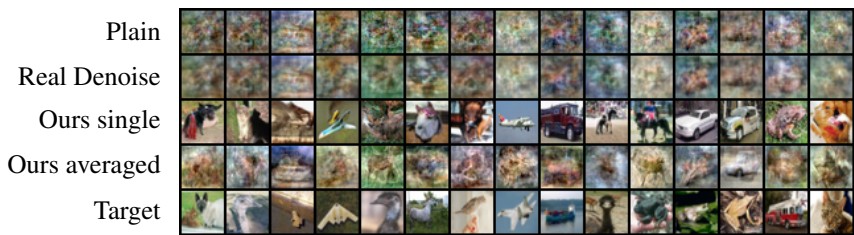

Figure 4: Illustration of the mitigation of hallucinations. Starting with the plain reconstructions in the top row (which do not appear to have any resemblance to any images from the training data), we apply different denoising/image restoration methods. The second row shows the result after applying the real image denoiser by Zamir et al. (2022). The middle row shows the image obtained after using our diffusion model once conditioned on the image in the top row. The fourth row shows the result after using our diffusion model 20 times conditioned on the image in the top row and averaging the results. While the diffusion model produces realistic images that are distinct from any of the images the network was trained on (the bottom row shows the closest match in terms of SSIM score), when running the diffusion model several times it becomes clear that the generation is not consistent. Averaging the results over 20 runs of the diffusion model produces images that no longer show a potentially misleading hallucination.

example, whether a diffusion model trained on data derived from Imagenet would perform similarly well when applied to CIFAR-10.

We also see an increased focus on types of training images that, so far, appear difficult to reconstruct as an interesting direction for future work. Images that largely show the sky or the sea, with only a relatively small and/or faint bird or boat occupying few of the total number of pixels cause significant challenges. First of all, these are difficult to reconstruct accurately and when the reconstructions are very noisy, our restoration procedure usually produces a similar mostly gray or blue picture, but fails to recover the accurate shape of the small original object in it. Secondly, measures such as SSIM are not well suited and tend to result in scores that are too optimistic for such images. The difference between a gray sky, with a small dark splash representing a plane or bird and a gray sky with no (or a slightly different) black splash is minor for measures such as SSIM while they can still feel significant to the human eye.

## 7 REPRODUCIBILITY STATEMENT

We will release the full dataset we created and view this is one of our central contributions. Due to its size of about 60 million images, we are sharing a subset of about 460,000 images as part of the supplementary materials for the reviewing period. In the supplementary materials, we are also providing pretrained weights for the diffusion model and for the BF-DnCNN model. We are sharing code on how to use the dataset to train these models and we are also sharing code on how the pretrained diffusion model can be used for the image restoration task. Further, we provide descriptions of the technical details of the dataset generation in section 2, of the training of the diffusion model section 3, and of the training of the BF-DnCNN model in section 4. The work by Haim et al. (2022) and their code is also publicly available and provides a reference point on how the reconstruction runs are performed and how the results are matched to training images using the SSIM metric.

## 8 ETHICS STATEMENT

The risk of the recovery of training data is not new, but is amplified by our work. So far, the method by Haim et al. (2022) has only been used to recover training data from relatively small models that are trained on relatively small datasets. It is currently computationally expensive to scale this up. However, until robust defenses are found and used, it is still advisable to treat the weights of a trained model in the same way as one would responsibly treat the data it was trained on.

We also note that, similar to membership inference attacks, training data reconstruction can also potentially be used by copyright holders to investigate whether their works have been used in the training of a model. See for example discussions in Carlini et al. (2023a).

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

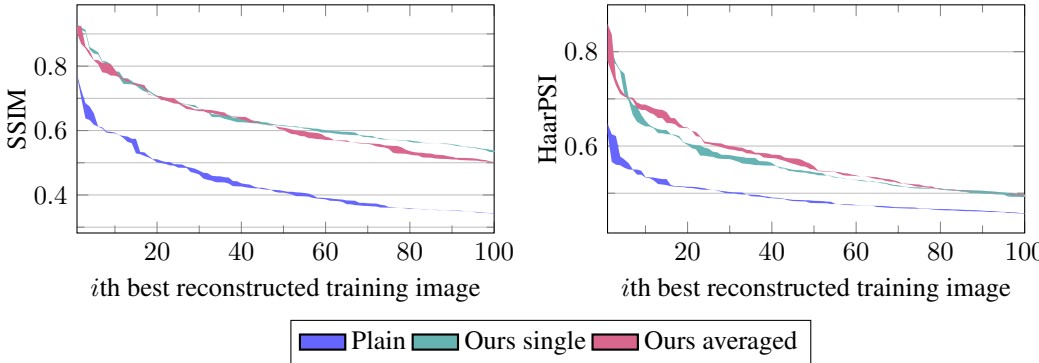

Figure 5: We randomly split our 274 successful reconstruction runs into two sets. We then generate plots as in Figure 2 for both sets and combine them to show the area between the smaller and the larger value of the two versions. To make the plot more readable, we do not show Deraining, Gaussian Denoise, Real Denoising, or BF-DnCNN. While the results are slightly worse overall than when we take the best reconstructions from all 274 combined, this plot shows that the results are very consistent when the reconstruction experiment is repeated.

## A   APPENDIX

### A.1   IMPACT OF THE RANDOM INITIALIZATION OF THE RECONSTRUCTION RUNS ON THE STABILITY OF RESULTS

In Figure 5, we demonstrate that the plots in Figure 2 are not overly affected by the randomness resulting from individual reconstruction runs.

### A.2   THE BEST RECONSTRUCTIONS FOR ALL 500 TRAINING IMAGES

The following figures in this appendix are similar to Figure 1 except that we show the best match obtained for each of the 500 training image for the different methods (instead of just 45 of them). Here images resulting from BF-DnCNN are also included. This order of the images is based on the similarity of the image in the fifth row compared to the target image in the bottom row (using the SSIM metric).[4]

---

[4]Where a black image is shown, the corresponding method did not result in any image which was most similar to the desired target image. In other words, all images were better matched to one of the other 499 training images.

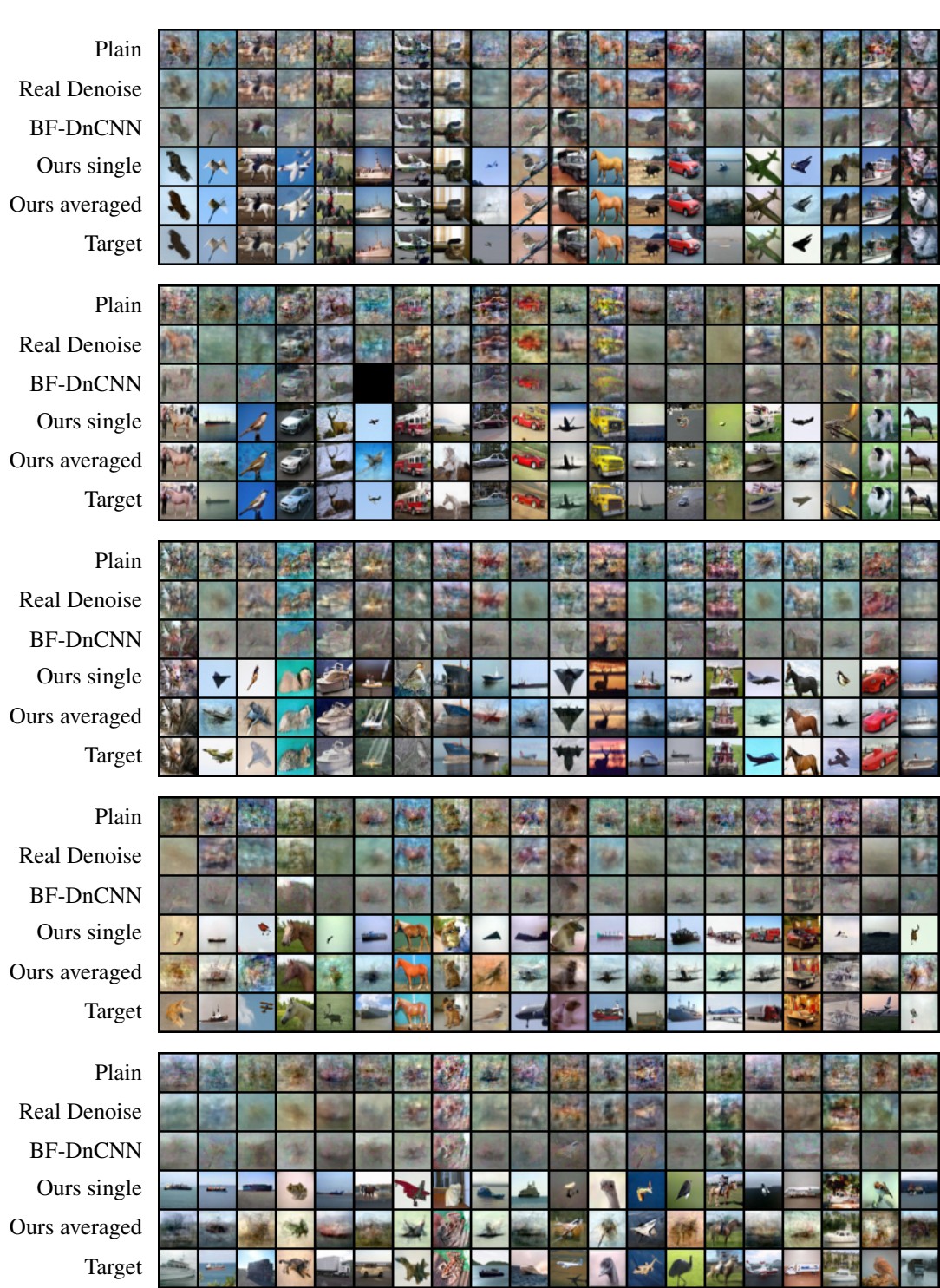

Figure 6: Reconstructions for target images 1–100.

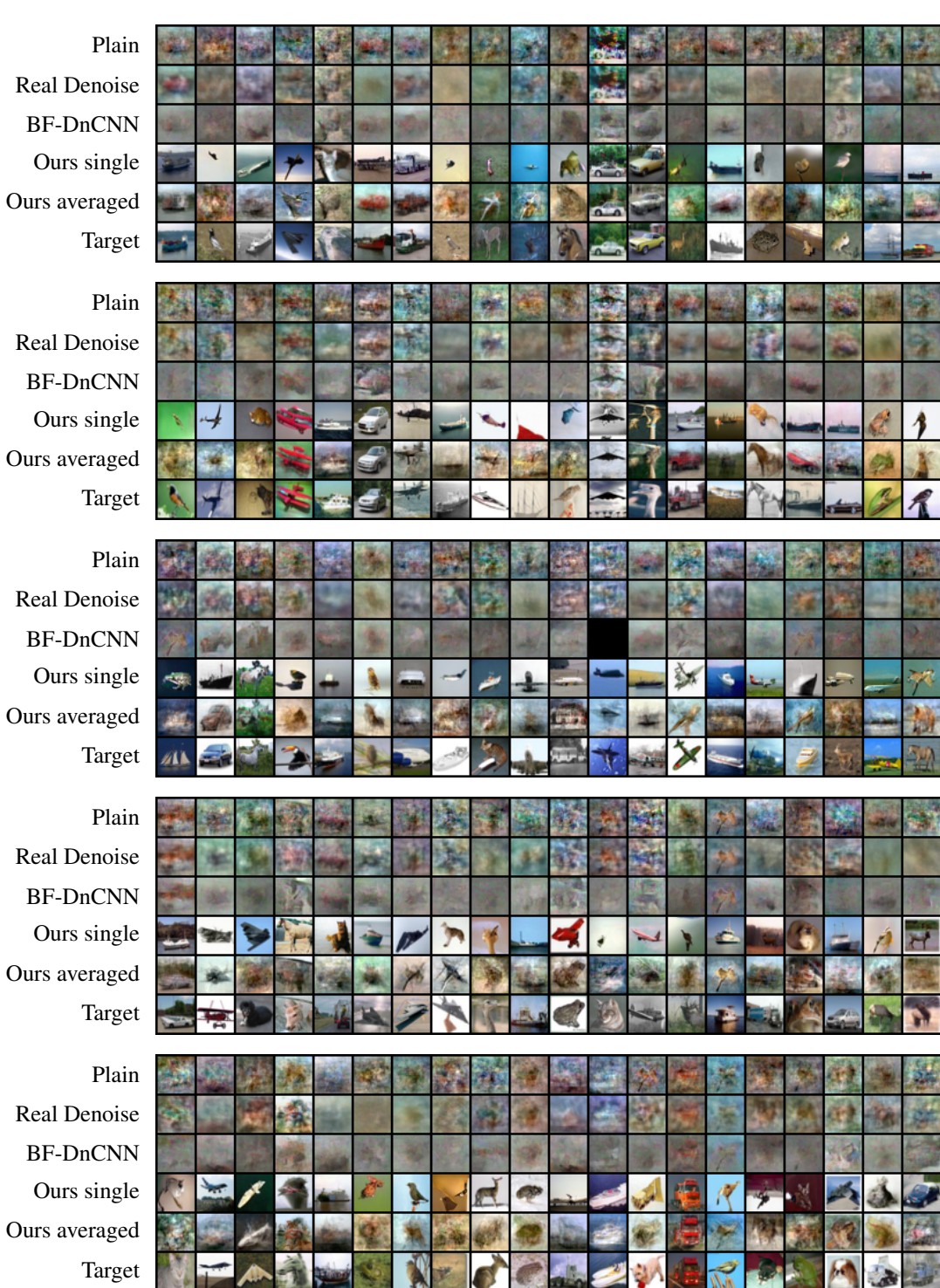

Figure 7: Reconstructions for target images 101–200.

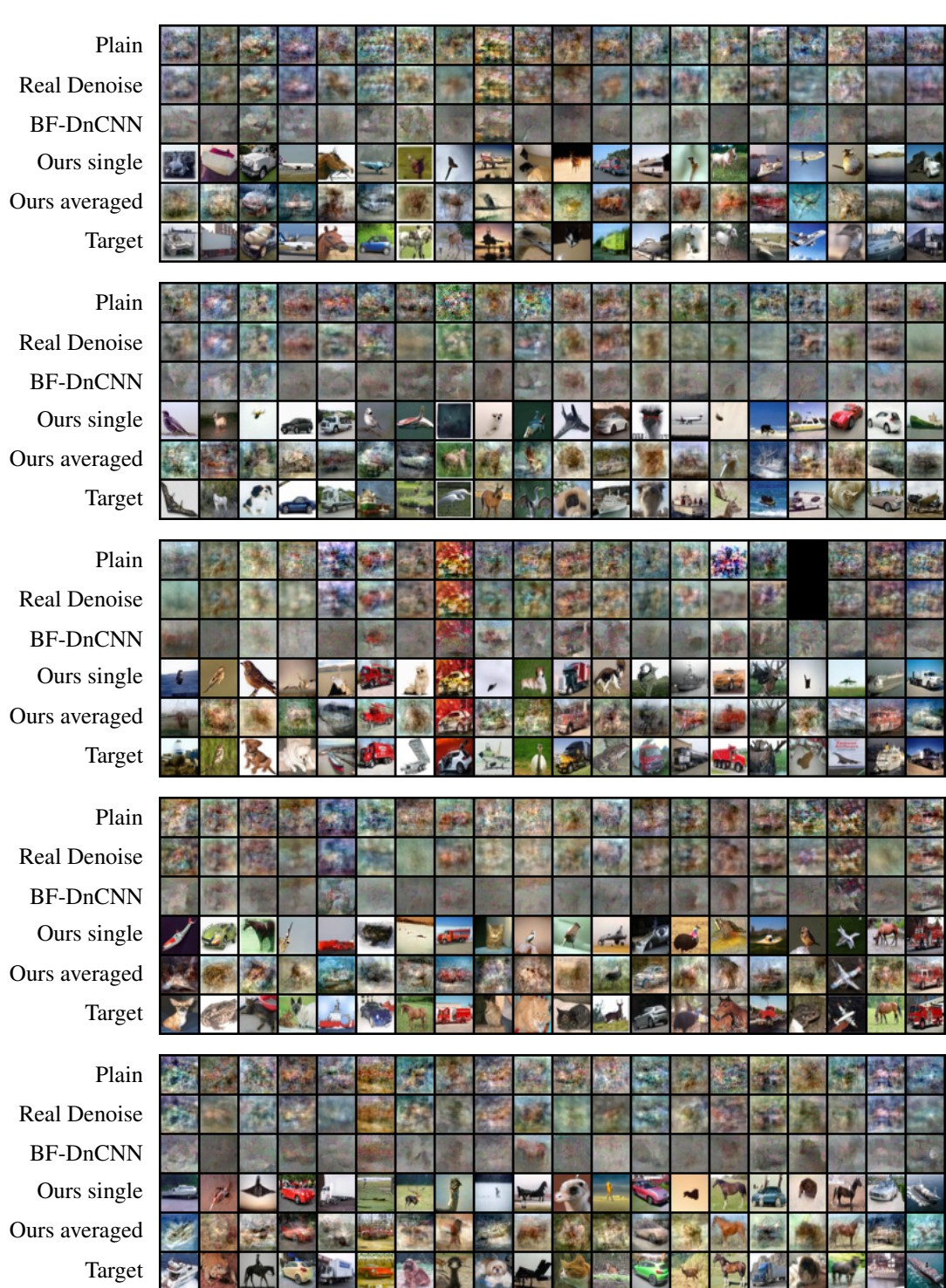

Figure 8: Reconstructions for target images 201–300.

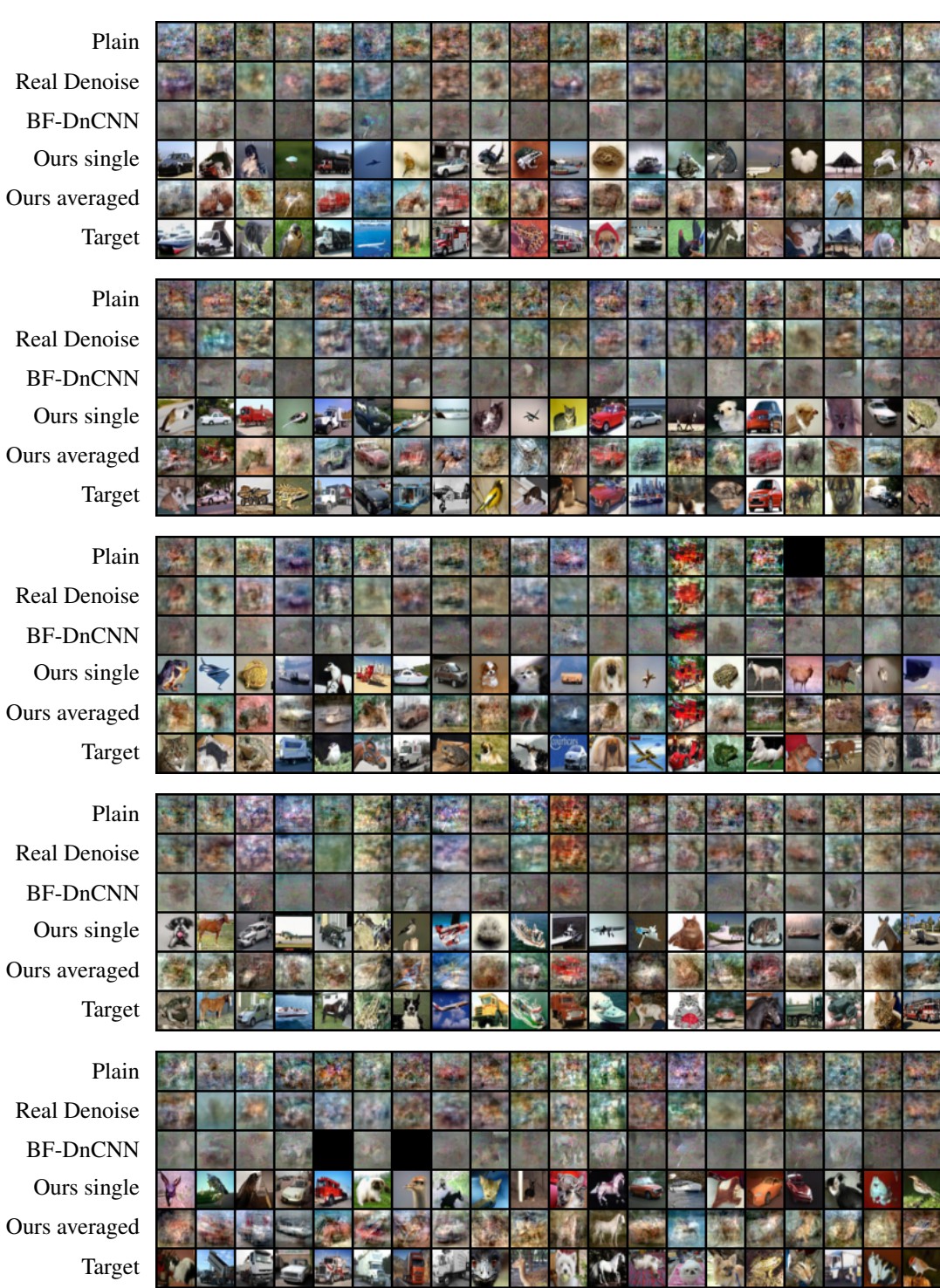

Figure 9: Reconstructions for target images 301–400.

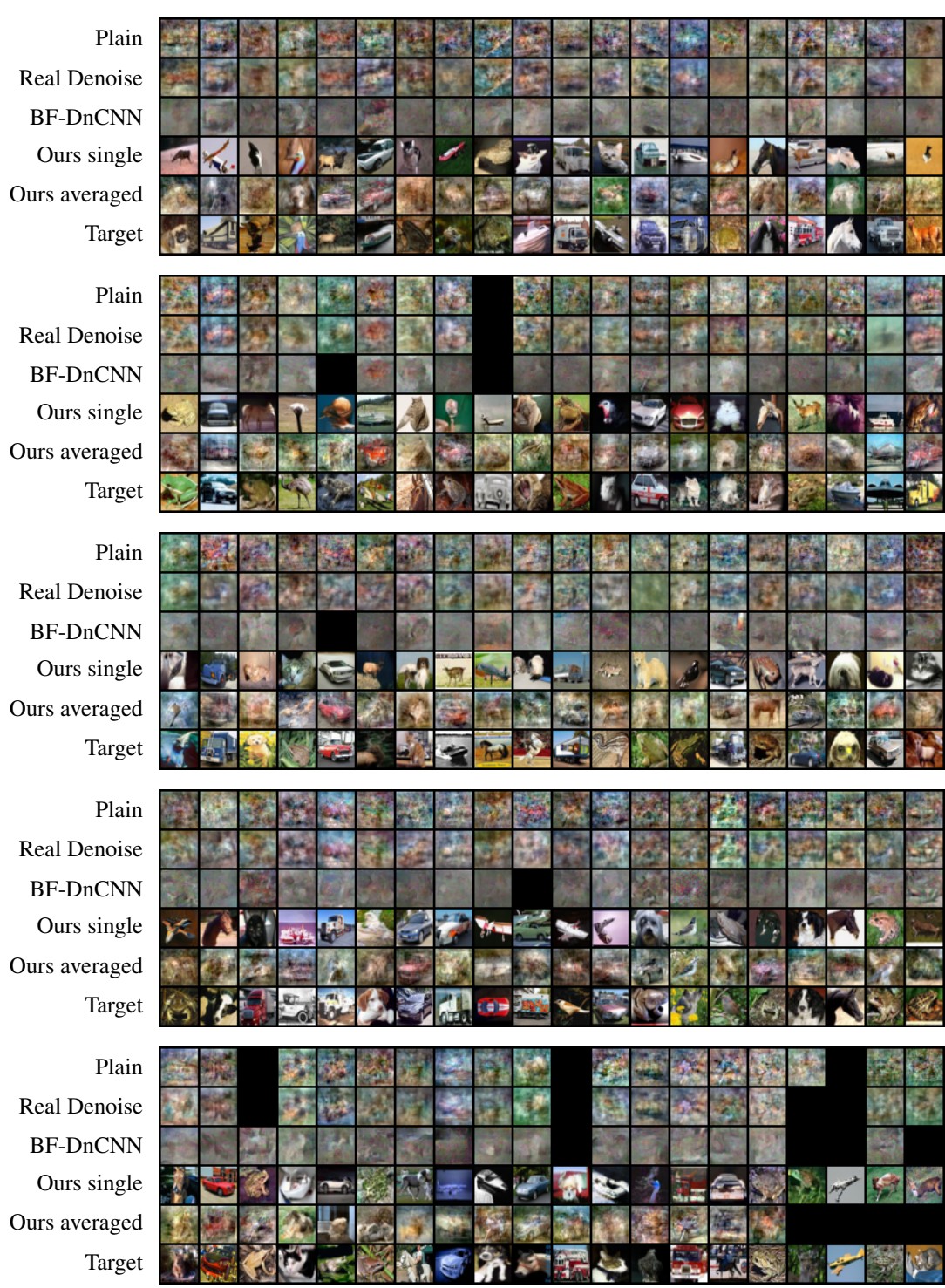

Figure 10: Reconstructions for target images 401–500.

