# OpenReview forum: "Image Restoration for Training Data Reconstructed from Trained Neural Networks"
_ICLR.cc/2025/Conference — ICLR 2025 Conference Withdrawn Submission_

### Official Review · Reviewer_XfiU · 2024-11-02

**Soundness:** 3
**Presentation:** 2
**Contribution:** 2
**Rating:** 5
**Confidence:** 2

**Summary:**

This paper introduces a novel approach to enhance the reconstruction of training data from trained neural networks, specifically targeting the noise and artifacts that often appear in reconstructed images when networks are trained on limited datasets. The authors propose an image restoration task using a diffusion model trained on a custom dataset of 60 million noisy reconstructions of CIFAR-10 images. This method demonstrates substantial improvement in the quality of reconstructed images, as evidenced by SSIM and HaarPSI scores.

**Strengths:**

1. The paper develop a novel approach to addressing noise and artifacts in reconstructed training data from neural networks. By formulating this issue as an image restoration task and applying a diffusion model, the authors achieve notable improvements in reconstruction quality.

2. The creation of a large-scale dataset with 60 million image pairs marks a substantial contribution to the research community, offering a valuable resource for advancing research in image restoration and the reconstruction of neural network training data.

3. The paper provides a comprehensive evaluation of the proposed method, demonstrating its effectiveness through quantitative metrics such as SSIM and HaarPSI scores. Additionally, the visual examples included effectively highlight the quality enhancements achieved by the method.

**Weaknesses:**

Actually, the field involved in this paper is not my area of expertise, so I find many parts of it confusing. However, I am doing my best to understand the content and offer my own insights where possible.

1. Generalization Across Datasets: While the method demonstrates strong results on CIFAR-10, its generalizability to other datasets with differing distributions remains uncertain. The paper could benefit from an exploration or discussion of the model's performance on a range of datasets to better assess its robustness.

2. Computational Cost: Generating the dataset and training the diffusion model is computationally intensive, which may limit accessibility for researchers with fewer resources. Addressing possible optimizations or providing guidelines for efficient implementation would enhance the approach’s practicality.

3. Model Complexity: The diffusion model is complex, potentially limiting its applicability in contexts where simpler or more interpretable models are preferred. A discussion on the trade-offs between model complexity and reconstruction quality would provide valuable insight into the model's versatility.

4. Evaluation Metrics: Although SSIM and HaarPSI are standard metrics, incorporating a wider range of evaluation metrics—particularly those capturing perceptual quality from a human perspective—would offer a more comprehensive evaluation of reconstruction quality and highlight practical improvements.

**Questions:**

Please see Weaknesses.

---

### Official Review · Reviewer_9Snd · 2024-11-03

**Soundness:** 1
**Presentation:** 1
**Contribution:** 2
**Rating:** 3
**Confidence:** 4

**Summary:**

In this paper, the authors aim to enhance the images reconstructed from the training data used to train the neural networks. The problem is important because reconstructing training data from trained neural networks has implications for data privacy leakage. This work is inspired by a previous work by Haim et al. The main contribution is to improve the image reconstruction quality. To achieve the goal, the authors first generated a large-scale dataset that consists of paired clean images and "noisy" images based on the method proposed by Haim et al. and the CIFAR10 dataset. Then the authors trained a diffusion model and a normal CNN to reconstruct the "noisy" images.

**Strengths:**

1. The targeted problem is quite an important one concerning data privacy.

**Weaknesses:**

1. There is a major flaw in the method: to train the diffusion model or the CNN to enhance the "noisy" images, the clean ground-truth images have been already used as the target in the loss function. In the practical training-data reconstruction attack proposed by Haim et al., the clean ground-truth images are not known or seen. Does this mean information leakage to the diffusion model or the CNN?

2. The contribution of this paper is too limited. The data simulation pipeline is mainly from the paper by Haim et al. Although the authors made some modifications to speed up the simulation, this is too trivial. The major contribution is using a diffusion model to enhance the reconstructed images, which as mentioned above has major flaws.

**Questions:**

1. The figures that visualize the reconstructed images should be enlarged for better visibility.

2. The authors only did experiments on CIFAR10. Can the method be generalized across different datasets, i.e., training on one dataset and validating on another dataset? This is important since it indicates an even more important problem about data privacy.

---

### Official Review · Reviewer_8zEy · 2024-11-03

**Soundness:** 1
**Presentation:** 3
**Contribution:** 1
**Rating:** 3
**Confidence:** 5

**Summary:**

This paper follows the method of Haim et al. 2022 to construct a dataset with clean and inversed  image (inversed image from classication neural networks) pairs. Then it trains a diffusion model  / a CNN model for image restoration. Although this paper is well-explained and well-written, I personally believe this type of work is like a highly engineered work that might not be suitable for ICLR.

**Strengths:**

1, well motivated, well explained and well-written

2, might be useful for the restoration of inversed images

**Weaknesses:**

Some quick comments are as below.

1, The dataset construction process is arguable. First, only some images of the CIFAR10 is used for training, rather than using the complete training set as typical classification networks. The degradation patterns might be different. Second, only the confidently matched ones are preserved. How about the examples that are hard to be inversed from the network?

2, Since the dataset is constructed from the “animals v.s. vehicles” task. All the pairs are either animals or vehicles. Can the trained model extend to other types of data for the subsequent restoration task?

**Questions:**

See weakness.

---

### Official Review · Reviewer_WGDX · 2024-11-07

**Soundness:** 2
**Presentation:** 2
**Contribution:** 2
**Rating:** 1
**Confidence:** 5

**Summary:**

This paper proposes a method to enhance image quality in reconstructions obtained through model inversion attacks. Building on prior work by Haim et al. (2022), which demonstrated methods for reconstructing training data from a model, the authors introduce a conditional diffusion model to remove the substantial noise and artifacts typically present in these reconstructions. By training the model on a large dataset of approximately 60 million CIFAR-10 images with artificially introduced noise and distortions, the method produces reconstructions that are visually closer to the original images. Evaluation metrics like SSIM and HaarPSI indicate that the restored images significantly improve in fidelity compared to both the noisy reconstructions and results obtained from general-purpose denoising tools.

**Strengths:**

1. By focusing on noise unique to model inversion, the approach goes beyond traditional denoising techniques.
2. The use of 60 million CIFAR-10 images enables effective training of a diffusion model tailored to this specific restoration task.

**Weaknesses:**

1. Limited Applicability: Primarily tested on CIFAR-10, this approach might struggle with natural or high-resolution images.
2. The method addresses only artifacts from specific reconstruction tasks, lacking versatility for general image restoration.
3. The experiments should be improved. Please refer to the details below.

**Questions:**

1. The primary limitation of this paper is its focus on CIFAR-10, a small, low-resolution dataset, which raises questions about the model's scalability to larger, higher-resolution datasets. While CIFAR-10 serves as an effective test case for early experiments, it is unclear whether this method can generalize to complex, high-resolution datasets like ImageNet.

2. Figure 1 only provides visual results, making it difficult to objectively assess the improvements claimed by the restoration model. Including quantitative metrics, such as SSIM and PSNR scores.

3. Figure 1 would benefit from a more comprehensive comparison that includes results from both classic image restoration methods, such as SwinIR, and advanced diffusion-based models like Stable Diffusion.

4. Although the paper claims to utilize a dataset of 60 million images, only a fraction of these images actually yield usable reconstructions due to the inherent limitations of the data generation process.

5. How sensitive is the restoration performance to different initialization parameters in the diffusion model?

---

### Note · Authors · 2024-11-20

**Comment:**

While some comments are based on misunderstandings, we generally find the feedback valuable and we thank the reviewers for their work and contributions.

**Withdrawal Confirmation:**

I have read and agree with the venue's withdrawal policy on behalf of myself and my co-authors.